# Mapping of Two New Rust Resistance Genes *Uvf-2* and *Uvf-3* in Faba Bean

**Usman Ijaz [1,2], Shimna Sudheesh [3], Sukhjiwan Kaur [3], Abdus Sadeque [1], Harbans Bariana [2,*], Urmil Bansal [2] and Kedar Adhikari [1]**

[1] Plant Breeding Institute, School of Life and Environmental Sciences, Faculty of Science, IA Watson Grains Research Centre, The University of Sydney, Narrabri, NSW 2390, Australia; usman.ijaz@agtbreeding.com.au (U.I.); abdus.sadeque@sydney.edu.au (A.S.); kedar.adhikari@sydney.edu.au (K.A.)

[2] Plant Breeding Institute, School of Life and Environmental Sciences, Faculty of Science, The University of Sydney, 107 Cobbitty Road, Cobbitty, NSW 2570, Australia; urmil.bansal@sydney.edu.au

[3] Department of Economic Development, Jobs, Transport and Resources, Agriculture Victoria Research Division, AgriBio, Centre for AgriBioscience, 5 Ring Road, Bundoora, VIC 3083, Australia; shimna.sudheesh@agriculture.vic.gov.au (S.S.); sukhjiwan.kaur@agriculture.vic.gov.au (S.K.)

\* Correspondence: harbans.bariana@sydney.edu.au

**Abstract:** Faba bean is gaining attention in Australia as a rotation grain legume where most of the country's produce is exported. Rust, caused by *Uromyces viciae-fabae*, is a major constraint to Faba bean production in eastern Australia and its chemical control results in increased cost of production. The deployment of diverse sources of resistance in new cultivars underpins economic and eco-friendly control of rust diseases of crops. A selection from cultivar Doza (Doza#12034) and a European accession Ac1655 exhibited seedling rust resistance against *U. viciae-fabae* pathotype 24–40. Doza#12034 and Ac1655 were crossed with a susceptible genotype Fiord and recombinant inbred line (RIL) $F_6$ populations were generated. Rust tests on Fiord/Doza#12034 and Fiord/Ac1655 $F_4$ and $F_6$ populations demonstrated monogenic inheritance of resistance in both crosses and the underlying resistance loci were named *Uvf-2* and *Uvf-3*, respectively. Genetic mapping of both RIL populations located *Uvf-2* and *Uvf-3* in chromosomes III and V, respectively. The SNPs that showed linkage with *Uvf-2* and *Uvf-3* were converted into kompititive allele specific PCR (KASP) assays. Markers *KASP_Vf_0703* and *KASP_C250539* flanked *Uvf-2* at 4.9 cM and 2.9 cM distances, whereas *Uvf-3* was flanked by *KASP_Ac×F165* (2.5 cM) and *KASP_vf_1090* (10.1 cM). Markers *KASP_Vf_0703* and *KASP_Ac×F165* can be used for marker-assisted selection of *Uvf-2* and *Uvf-3*, respectively, after confirming parental polymorphisms.

**Keywords:** faba bean; rust resistance; molecular mapping; marker-assisted selection

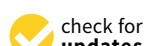

## 1. Introduction

Faba bean (*Vicia faba* L.) is an early domesticated crop species and a source of protein for humans and animals. It is known to have originated from three primary centers in the Fertile Crescent; Near East, Iraq and Iran with unknown wild progenitor(s) and all hybridization efforts with other *Vicia* species have proved futile [1–3]. In the absence of wild relatives of modern faba bean, genetic diversity is restricted to the available primary gene pool.

China, Ethiopia and Australia contribute more than 70% to the global faba bean production [4]. It was introduced in Australia in the late 20th century as a rotational crop after wheat and since been gradually established as an important legume crop because of its increasing export potential [5]. Australia is the world's leading exporter of faba bean with an average of 218 thousand tons (69% of total produce) to Egypt and other Middle Eastern countries [6]. It plays a key role in soil fertility restoration through nitrogen fixation

by its successful association with *Rhizobium* sp. (root nodulation bacterium), therefore making a significant contribution to agricultural sustainability [7].

The key objectives in faba bean improvement programs include resistance to biotic and abiotic factors, matching phenology to targeted cropping regions and incorporation of better agronomic traits for higher yield [8,9]. In Australia, faba bean production is threatened by fungal and viral diseases. Faba bean rust, caused by *Uromyces viciae-fabae*, is the most commonly occurring disease in major cultivation sites of northern New South Wales (NSW) and southern Queensland (QLD) [10]. Rust can appear throughout the crop season, but early onset of the disease can destroy 70–80% of the crop [11,12]. Unlike cereals, none of the faba bean genotypes is immune to rust [5,13]. However, a late necrosis observed in some cases [13,14] can reduce disease severity.

Sustained release of rust resistant cultivars of faba bean relies on the availability of genetically diverse sources of resistance. No wild species is sufficiently close to allow inter-specific gene transfer, which restricts genetic diversity to the primary gene pool [15]. Pioneering work on exploring inheritance of rust resistance dates back to the 1980s when Canadian researchers identified and named three to seven resistance genes in various studies [16–18]. A similar set of Canadian genotypes showed resistance when tested under field and greenhouse conditions in Spain [9,19] and Australia [5,13]. Two independently inherited dominant rust resistance genes with possible additivity, and a complimentary gene pair were reported against Australian faba bean rust isolates [13,20]. Although various sources of rust resistance were reported, none of these were precisely characterised and genetically mapped. Only *Uvf-1* was mapped using Randomly Amplified Polymorphic DNA (RAPD) markers [21] and information on the use of this RAPD marker in faba bean breeding is lacking.

Genetic mapping leads to relative ordering of various loci on chromosomes of an organism determined through recombination [22]. It is a powerful tool that facilitates the development of closely linked DNA markers with a particular phenotype/locus [23]. The application of trait-linked markers to improve selection efficiency in breeding programs is highly desirable to avoid expensive and time-consuming phenotypic tests, particularly in the case of quantitatively inherited traits. Simple sequence repeat (SSR) markers have been and are still being used for mapping of economic traits and their marker-assisted selection (MAS) in many crop plants. Genotyping-by-sequencing based on transcriptome (GBS-t) has maximized the capacity of marker identification in plants species with large genome sizes in the absence of whole genome sequences [24]. Therefore, it has become the method of choice for marker development in legume crops such as field pea [25], yellow lupin (*Lupinus luteus* L.) [26], lentil [27], chickpea (*Cicer arientinum* L.) [28], and faba bean [29,30]. GBS-t enables the discovery of trait-linked SNP markers which are co-dominant in nature and amenable to high throughput systems for selection in crop improvement programs [31].

The moderately rust resistant cultivar Doza was released in 2008 for cultivation in northern NSW and southern Queensland regions [6]. Doza was heterogenous for rust response at the time of release. Adhikari et al. [13] reported three single rust resistant plants (#12034, #12035 and #14916), selected by Mr. J. van Leur, under field conditions. Doza#12034 expressed strong chlorosis leading to the death of infected leaves on the first and the second nodes within 2–3 days post-sporulation in the seedling tests [20]. The faba bean accession Ac1655 was reported to exhibit IT 2$^+$ in greenhouse against rust isolates collected from Cordoba, Spain [14].

The objectives of this study were to: (1) characterise seedling rust resistance in Fiord/Doza#12034 and Fiord/Ac1655 recombinant inbred line (RIL) populations, (2) determine genomic locations of resistance in Doza#12034 and Ac1655 through molecular mapping, (3) develop KASP markers linked with the underlying resistance loci and (4) validate resistance linked markers on a set of local and exotic cultivars/genotypes to demonstrate their utility in marker-assisted breeding.

## 2. Materials and Methods

### 2.1. Plant Material and Development of RIL Populations

Two rust resistant faba bean genotypes, a single plant selection #12034 from an Australian faba bean cultivar Doza and the European line Ac1655 (V-300) from Spain (originally from central Europe), were selected for the present study. The RIL populations were developed by crossing resistant parents Doza#12034 and Ac1655 with the susceptible cultivar Fiord at the University of Sydney Plant Breeding Institute, Narrabri under controlled conditions to avoid cross pollination. One hundred-forty and 145 $F_2$ seeds were harvested from Doza#12034/Fiord and Ac1655/Fiord crosses, respectively. A single pod from each $F_2$ plant was harvested and used for generation advancement. This process was repeated for four generations and finally $F_6$ population of 104 Fiord/Doza#12034 and 120 Fiord/Ac1655 individuals were harvested. Both resistant parents, Doza#12034 and Ac1655, were also crossed with each other to study allelic relationship of the underlying genes and to understand their interaction. From this cross, 92 $F_2$ seeds were harvested and no further population advancement was carried out.

### 2.2. Pathogen Material and Inheritance Studies

The rust isolate-8 collected from Breeza, New South Wales, named pathotype 24–40 [32], was used to study inheritance of rust resistance in Fiord/Doza#12034 and Fiord/ Ac1655 $F_4$ and $F_6$ RILs and Doza#12034/Ac1655 $F_2$ population under controlled environment conditions at the University of Sydney Plant Breeding Institute, Cobbitty. Parents Doza#12034, Ac1655 and Fiord were planted as controls with each experiment. Ten seeds of each RIL and five seeds of each parent were sown under greenhouse conditions and inoculated according to the procedure described by Adhikari et al. [13]. The wheat rust infection type (0–4) scale was used, where 0 = no visible uredinia, ; = hypersensitive pin head flecks, 1 = small uredinia with necrosis, 2 = small to medium uredinia with by green island and necrosis/chlorosis, 3 = medium sized uredinia with or without chlorosis and 4 = large uredinia without chlorosis. The $F_4$ populations were classified as homozygous resistant (HR), segregating (SEG) and homozygous susceptible (HS), whereas $F_6$ RIL populations were categorized as HR and HS. Chi-squared ($\chi^2$) analyses were performed to determine the number of genes involved in conditioning rust resistance in Doza#12034 and Ac1655.

### 2.3. DNA and RNA Extraction

DNA was extracted from seedlings of Fiord/Doza#12034 and Fiord/Ac1655 RILs, parents (Doza#12034, Ac1655 and Fiord) and a set of 40 genotypes (Supplementary Table S1) using modified Cetyltrimethylammonium bromide (CTAB) method [33] with an addition of 2% Polyvinylpyrrolidone (PVP) in extraction buffer. The quality of DNA was analysed on 1% agarose gel, quantification was performed using Nanodrop spectrophotometer (Nanodrop Technologies) and working dilutions of 30 ng/μL concentration were prepared for marker genotyping. All DNA samples were stored at −20 °C.

Fiord/Doza#12034 ($F_6$) and Fiord/Ac1655 ($F_4$) populations were sown in a tray (54 cm × 28 cm) and leaf tissue from each line (two replicates) was harvested and immediately transferred to liquid nitrogen before storage at −80 °C. Total RNA was extracted from all samples using QIAGEN RNeasy 96 kit following manufacturer's instructions.

### 2.4. Genotyping-by-Sequencing and SNP Discovery

Fiord/Doza#12034 and Fiord/Ac1655 $F_4$ populations were genotyped with Illumina Infinium 1536 -SNP array. In addition, RNAseq-based genotyping-by-sequencing (GBS-t) was performed on Fiord/Ac1655 $F_4$ and Fiord/Doza#12034 $F_6$ RIL populations.

Following mRNA enrichment in both populations, the sequencing library preparation was performed using the SureSelect stranded RNA library preparation kit (Agilent Technologies, Santa Clara, CA, USA) following the manufacturer's instructions. Libraries were evaluated using the TapeStation 2200 platform (Agilent Technologies, Santa Clara, United States), pooled, and quantified using the KAPA library quantification kit (KAPA

Biosystems, Wilmington, NC, USA). Sequencing data were generated using Illumina HiSeq 3000 (Illumina Inc., San Diego, CA, USA). The reference transcriptome of Doza published by Braich et al. [34] was used for SNP discovery, whereas a less comprehensive sequence was used in the case of Ac1655.

### 2.5. Construction of Linkage Map

The SNP alleles were named as 'A': Doza or Ac1655 (in respective population), 'B': Fiord, 'H': heterozygote and '-' for missing values. Genotypic data were tested for segregation distortion of markers using $\chi^2$ test and markers with distorted segregation were removed from further analysis. Monomorphic markers, redundant markers and markers with more than 10% missing data were also excluded. Mapping was performed using Kosambi [35] mapping function with Map Manager version QTXb20 [36] for the Fiord/Doza#12034 RIL population. The linkage groups (LGs) were generated using the R/ASMap package [37] and linkage maps and the joint linkage map diagrams were drawn using the MapChart software [38].

### 2.6. KASP Genotyping

Rust resistance-linked SNPs from both RIL populations were converted into Kompetitive Allele Specific PCR (KASP) assays. The primers were designed using online tool BatchPrimer3 [39]. The primers with GC content ~50% were picked to construct allele specific-1(A-1), allele specific-2 (A-2), and common (C) primers (Supplementary Table S2). The KASP primers were assayed on the entire respective RIL population and parents using KASP technology (LGC genomics, UK) described in Nsabiyera et al. [40]. The closely linked KASP markers were also assayed on a set of 40 genotypes to test polymorphism (Supplementary Table S1).

## 3. Results

### 3.1. Genetic Analysis

The resistant parents Doza#12034 and Ac1655 produced infection types (ITs) 1C and 12C, respectively, and the susceptible parent Fiord expressed IT 33$^+$, when tested against rust pathotype 24-40 (Figure 1). Screening of $F_4$ generations of both crosses showed monogenic segregation for rust resistance. Forty-four Fiord/Doza#12034 $F_4$ lines were scored HR, 24 SEG and 42 HS ($\chi^2$ $_{3:2:3}$ = 0.64, non-significant at $p$ = 0.05 and 2 d.f.). Similarly, the Fiord/Ac1655 $F_4$ population included 34 HR and 17 SEG and 32 HS lines ($\chi^2$ $_{3:2:3}$ = 0.97, non- significant at $p$ = 0.05 and 2 d.f.). Monogenic segregation for rust resistance was confirmed through tests on $F_6$ generations of both crosses (Table 1).

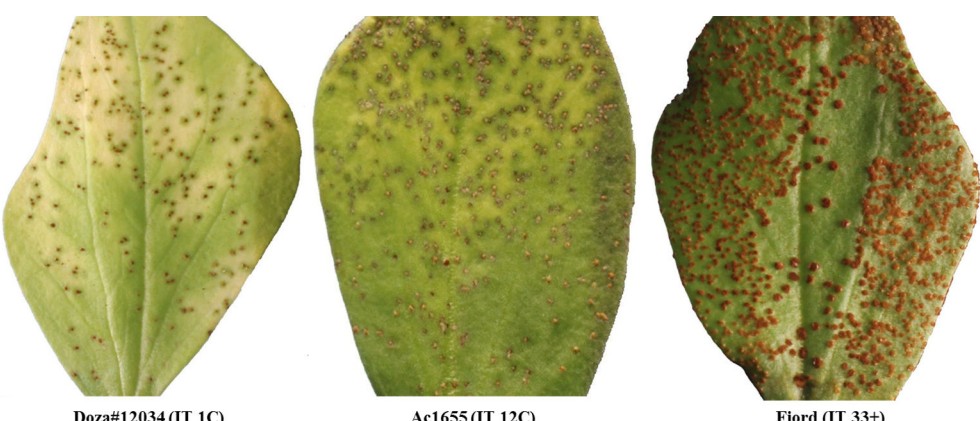

**Figure 1.** Infection types produced by parental genotypes when inoculated with the *U. viciae-fabae* pathotype 24–40.

**Table 1.** Seedling rust response variation among Fiord/Doza#12034, Fiord/Ac1655 RIL, and Doza#12034/Ac1655 populations when tested with *Uromyces viciae-fabae* pathotype 20–40 under greenhouse conditions.

| Phenotype Classes (IT) | Number of Families | | |
|---|---|---|---|
| | Observed | Expected | |
| Fiord/Doza#12034 RIL | | | $\chi^2_{(1:1)}$ |
| 1C | 57 | 52 | 0.48 |
| $33^+$ | 47 | 52 | 0.48 |
| Total | 104 | 104 | 0.96 |
| Fiord/Ac1655 RIL | | | $\chi^2_{(1:1)}$ |
| 12C | 69 | 60 | 1.35 |
| $33^+$ | 51 | 60 | 1.35 |
| Total | 120 | 120 | 2.70 |
| Doza#12034/Ac1655 F$_2$ | | | $\chi^2_{(9:3:3:1)}$ |
| ;1$^=$ | 46 | 51.75 | 0.64 |
| 1C | 23 | 17.25 | 1.92 |
| 12C | 19 | 17.25 | 0.18 |
| $33^+$ | 4 | 5.75 | 0.53 |
| Total | 92 | 92 | 3.27 |

Table value of $\chi^2$ at $p = 0.05$, 1 *df* = 3.84, 3 *df* = 7.81.

The F$_1$ seedlings of Doza#12034/Ac1655 cross produced IT ;1$^=$. The F$_2$ population showed digenic segregation (15 resistant: 1 susceptible) to demonstrate that the two sources of resistance carry genetically independent genes (Table 1). The ITs among resistant plants varied from ;1= to 12C. These results confirmed the independent and dominant inheritance of resistance in both parents. The rust resistance loci from Doza#12034 and Ac1655 were named *Uvf-2* and *Uvf-3*, respectively.

*3.2. Molecular Mapping of Rust Resistance*

3.2.1. Construction of Linkage Maps

A total of 409 and 557 markers were used to generate linkage groups for Fiord/Doza#12034 and Fiord/Ac1655 F$_4$ populations using JoinMap 4 software. The Fiord/Doza#12034 linkage map covered 969 cM and the Fiord/Ac1655 spanned 1434 cM. QTL detection was performed using simple interval mapping (SIM) and composite interval mapping (CIM).

3.2.2. Mapping of Uvf-2

Single genomic region on chromosome III of Fiord/Doza#12034 showed association with rust resistance (Figure 2a) and it was named q_rust_Doza. It was flanked by SNP markers *Vf_0685* and *Vf_0035*. Six SNPs were converted into KASP assays. Four KASP markers (*KASP_Vf_0703*, *KASP_Vf_0493*, *KASP_Vf_0198* and *KASP_Vf_0032*) that showed polymorphisms between parents Doza#12034 and Fiord were mapped on 104 lines of the Fiord/Doza#12034 F$_6$ RIL population and covered a 37.1 cM interval. Rust resistance gene *Uvf-2* mapped 4.9 cM away from the marker *KASP_Vf_0703* (Figure 2b).

A total of 15,672 high-confidence SNPs were identified by alignment of GBS-t reads against the reference transcriptome data of Doza. Of those, 20% (3138) showed distorted segregation and 7988 had more than 10% missing data. These markers were excluded from the analysis. Redundant markers were filtered out from the remaining 4546 polymorphic SNPs resulting in 3924 that were used for constructing the genetic linkage map. In total, 2784 (out of 3924) SNP markers were distributed across 12 linkage groups (LGs) covering 1026.59 cM with an average marker density of 0.369 cM (Supplementary Table S3 and Supplementary Figure S1) and 1140 SNPs remained unlinked.

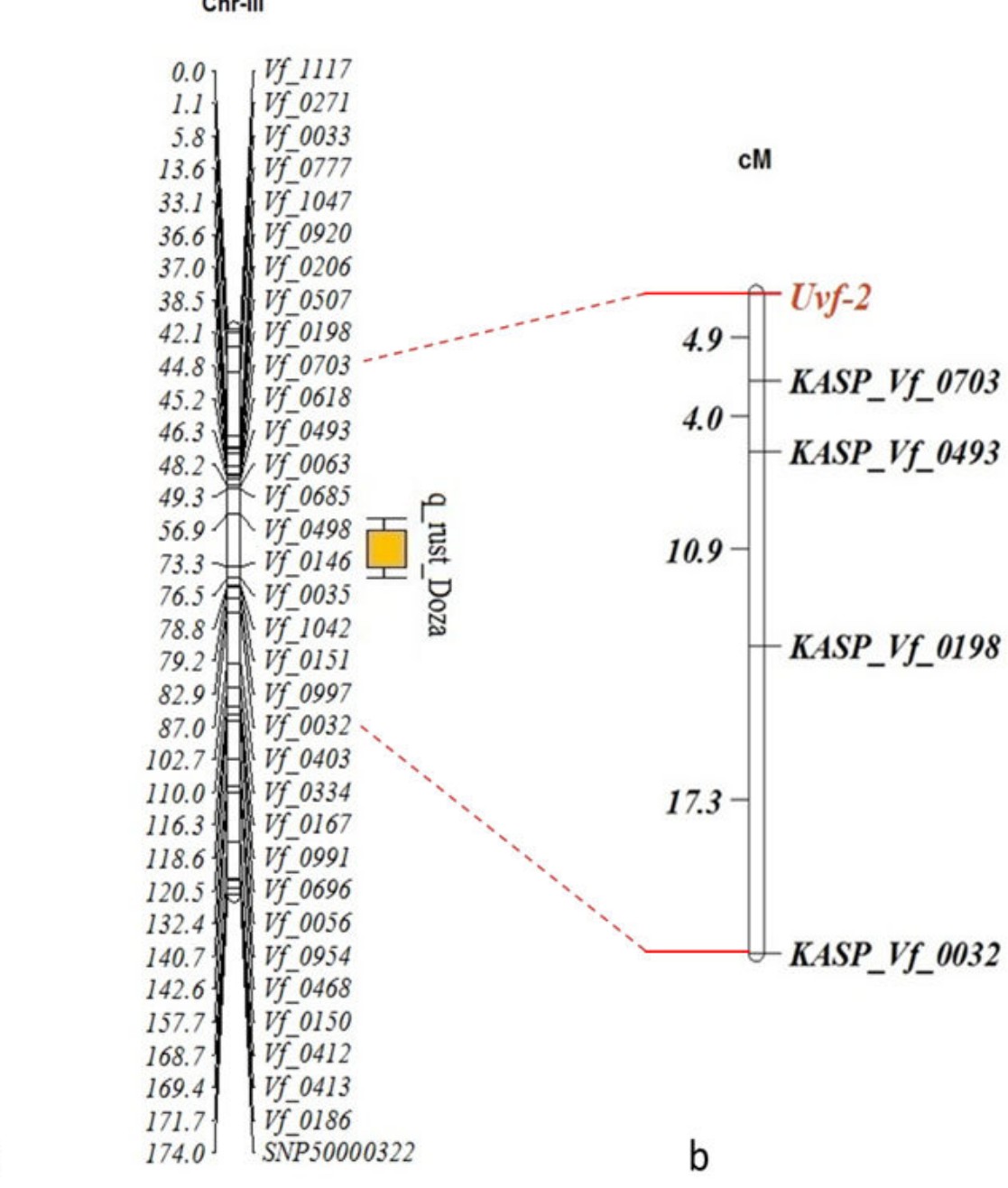

**Figure 2.** Linkage maps of chromosome III of the Fiord/Doza#12034 RIL population: (**a**) location of QTL for rust resistance in F₄ and (**b**) association of KASP markers converted from SNP in the QTL region with rust resistance gene *Uvf-2*.

In GBS-t data, 14 SNPs showed strong association with rust response of 88 Fiord/Doza#12034 RILs. Of these 14 SNPs, seven (*C250539, C241629, C150949, C244859, C208897, C246203* and *C229467*) co-segregated and mapped at 1.4 cM away from *Uvf-2* and the remaining seven SNPs (*C198227, C233649, C225873, scaffold3124, C241205, C174045* and *C239085*) mapped at 0.7 cM from this cluster (Figure 3a). The KASP markers designed from these 14 SNPs (Supplementary Table S2) were genotyped on 104 Fiord/Doza#12034 RILs. The marker *KASP_C250539* mapped closest to *Uvf-2* at a distance of 2.9 cM (Figure 3b). The joint linkage map was also constructed by combining flanking KASPs developed from GBS-t (Figure 3b) and Illumina array (Figure 2b) to refine the map location of rust resistance gene *Uvf-2* (Figure 3c). Markers *KASP_Vf_0703* and *KASP_C250539* flanked *Uvf-2*.

### 3.2.3. Mapping of Uvf-3

QTL analysis of rust resistance showed association of markers from chromosome V with rust resistance in the Fiord/Ac1655 F$_4$ population and it was designated q_rust_Ac1655 (Figure 4a). It was delimited by markers *AC×F138* and *Vf_1318*. Twenty SNP markers (*Vf_0096*, *Vf_0110*, *Vf_0565*, *AC×F213*, *AC×F40*, *AC×F260*, *AC×F275*, *AC×F165*, *Vf_0592*, *Vf_1319*, *AC×F140*, *AC×135*, *VF_1090*, *AC×F138*, *VF_1318*, *Vf_0257*, *Vf_023*, *AC×F157*, *AC×F327* and *Vf_0457*) from the QTL interval were converted into KASP assays (Supplementary Table S2). Three KASP markers (*KASP_AC×F165*, *KASP_Vf_1090* and *KASP_Vf_0203*) showed polymorphisms between parents and were assayed on the entire Fiord/Ac1655 F$_6$ RIL population. These markers resulted in a genetic map with resistance gene *Uvf-3* spanning across 20 cM (Figure 4b). The KASP markers *KASP_AC×F165* and *KASP_Vf_1090* flanked *Uvf-3* at 2.5 cM 10.1 cM, respectively (Figure 4b).

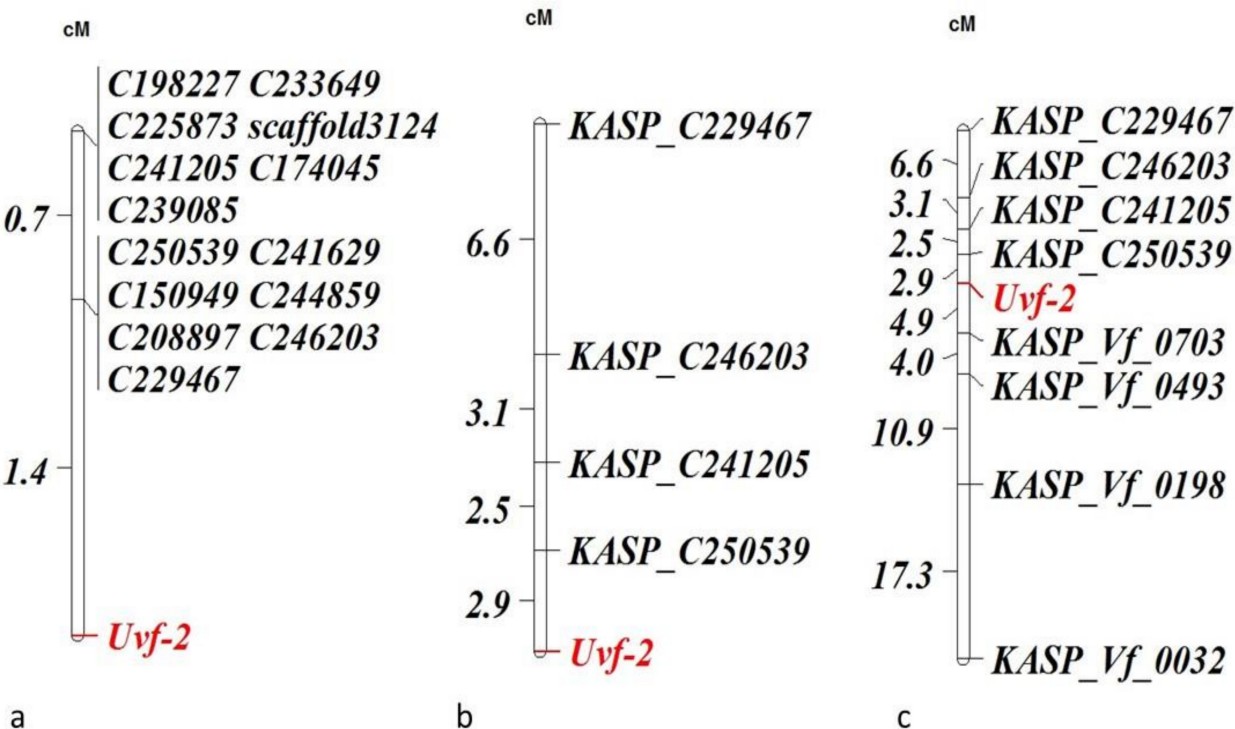

**Figure 3.** Linkage maps of chromosome III of Fiord/Doza#12034 population showing association of markers linked with rust resistance gene *Uvf-2*: (**a**) constructed from SNPs identified through GBS-t (88 RILs), (**b**) based on KASP markers developed for SNPs detected from GBS-t and (**c**) joint linkage map constructed with KASPs derived from 1536 Illumina and GBS-t.

### 3.3. Validation of KASP Markers

#### 3.3.1. Uvf-2

The flanking markers *KASP_Vf_0703* and *KASP_C250539*, which showed close linkage with *Uvf-2* were evaluated on a set of 40 genotypes (Table 2). The KASP markers *KASP_Vf_0703* and *KASP_C250539* amplified the *Uvf-2* linked alleles *G:G* and *A:A*, respectively, in resistant parent (Doza#12034). *KASP_Vf_0703* amplified the Fiord allele (*A:A*) in all the tested genotypes, except Doza#14916, which is another selection from cultivar Doza. Similarly, *KASP_C250539* also produced resistance linked allele in Doza#14916, but it amplified resistance linked allele in genotypes known not to carry *Uvf-2* (PBA Warda, IX585c/1-11 and IX524Rb-2-1), indicating loose genetic association with resistance. These results indicated that *KASP_Vf_0703* predicted the presence of *Uvf*-2 accurately among the tested genotypes.

### 3.3.2. Uvf-3

The closely linked marker *KASP_AC×F165* (2.5 cM) produced '*C:C*' allele in resistant parent Ac1655 and '*T:T*' allele in susceptible parent Fiord. Four cultivars, 10 breeding lines, four field selections, and nine germplasm accessions produced the '*T:T*' allele similar to the susceptible parent Fiord (Table 2).

This marker amplified the *Uvf-3*-linked allele '*C:C*' in genotypes (IX474/4-3, IX561f-4-2, IX552Rb-2-4, IX553Rc-2-4, AF10089 Ac1231#14905, Acc740, Ac1866#15013) known not to possess this resistance locus. In addition, three cultivars Doza, PBA Warda and PBA Nasma were mixed ('*T:C*' allele). *KASP_AC×F165* marker was polymorphic in 32 out of 40 genotypes (80%) tested in this study. We recommend the use of *KASP_AC×F165* for marker-assisted selection of *Uvf-3* after testing the parental polymorphism.

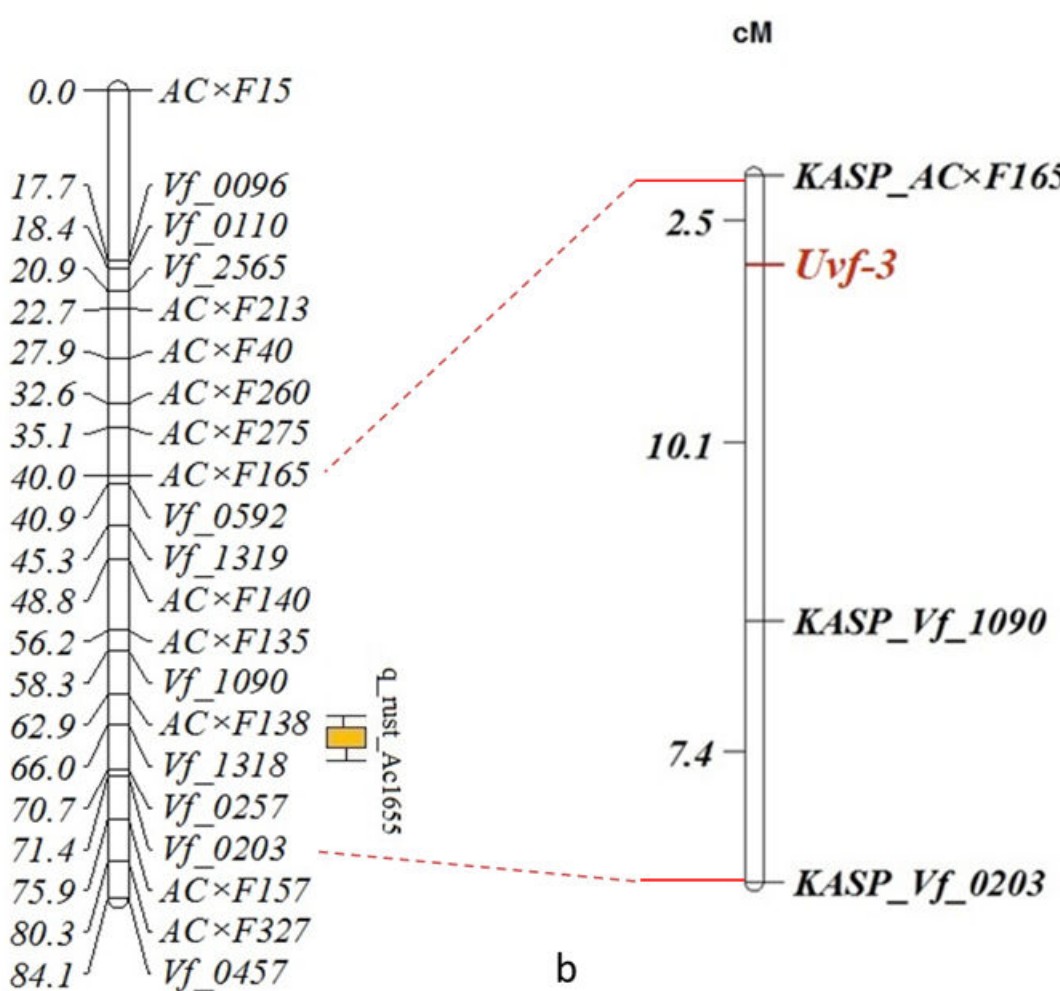

**Figure 4.** Linkage maps of chromosome V of the Fiord/Ac1655 RIL population: (**a**) location of QTL for rust resistance in 1536 Illumina and GBS-t based map and (**b**) association of KASP markers converted from SNP in the QTL region with *Uvf-3*.

**Table 2.** Validation of *Uvf-2* and *Uvf-3*-linked KASP markers on a set of local and exotic cultivars/genotypes.

| *Uvf-2* | *KASP_Vf_0703* |
|---|---|
| *Parents* | |
| Doza#12034 | *G:G* |
| Fiord | *A:A* |
| *Cultivars* | |
| Ascot, Doza, Icarus, PBA-Rana, PBA-Nasma | *A:A* |
| PBA-Cairo | |
| PBA-Warda | |
| *Australian breeding lines* | |
| 11NF001a-6, 11NF014b-1, IX486/7-6, IX561f-4-2, IX552Rb-2-4, IX525C-1-10, IX553Rc-2-4, AF10089, AF03109, AF09167, AF08161, IX114#15033, IX585c/1-11, IX474/4-3, IX524Rb-2-1 | *A:A* |
| *Australian field selections* | |
| Det-2, Cairo seln#7 | *A:A* |
| Doza#14916 | *G:G* |
| *Germplasm accessions* | |
| Ac1655, Ac1227#14908, Ac1221#14910, Ac1594#5004, Ac1228#14902, Ac1231#14905, Acc740, Ac1866#15013, Ac1257#14904, Ac0973#4902, Ac1206#4919, Ac1269#8127, Ac0805#4912 | *A:A* |
| *Uvf-3* | *KASP_Ac×F165* |
| *Parents* | |
| Ac1655 | *C:C* |
| Fiord | *T:T* |
| *Cultivars* | |
| Ascot, Cairo, Icarus, PBA Rana | *T:T* |
| Doza, PBA Warda, PBANasma | *T:C* |
| *Australian breeding lines* | |
| 11NF001a-6, 11NF014b-1, IX486/7-6, IX524Rb-2-1, IX114#15033, IX525C-1-10, IX585c/1-11, AF03109, AF09167, AF08161 | *T:T* |
| IX474/4-3, IX561f-4-2, IX552Rb-2-4, IX553Rc-2-4, AF10089 | *C:C* |
| *Australian field selections* | |
| Det-2, Doza#12034, Doza#14916, Cairo seln#7 | *T:T* |
| *Germplasm accessions* | |
| Ac1227#14908, Ac1221#14910, Ac1594#5004, Ac1228#14902, Ac1257#14904, Ac0973#4902, Ac1206#4919, Ac1269#8127, Ac0805#4912 | *T:T* |
| Ac1231#14905, Acc740, Ac1866#15013 | *C:C* |

## 4. Discussion

The dawn of next generation sequencing (NGS) technologies has enabled plant scientists to generate transcriptome sequence assemblies to study crops where whole genome sequences are unavailable and faba bean represents this category [34,41–43]. Although QTL for several desired economic traits of faba bean have been reported [30,44], markers linked with rust resistance did not get attention. Faba bean rust is an important disease and is commercially managed by spraying fungicides. Breeding rust resistant cultivars is the best disease management strategy where success relies on the identification of new resistance sources and their deployment into high yielding and well adapted agronomic backgrounds [20]. It is difficult to ascertain the presence of more than one gene using conventional selection technologies and the development of molecular markers closely linked with disease resistance genes can overcome this hurdle [45].

Monogenic inheritance of rust resistance in Doza#12034 and Ac1655 was demonstrated. Adhikari et al. [13] also reported the presence of single resistance genes in another Doza selection (#12035) and Ac1655. These inheritance studies further confirmed the previous findings by Adhikari et al. [13] and permanent names *Uvf-2* and *Uvf-3* were assigned to resistance loci present in Doza#12034 and Ac1655, respectively. It was also concluded that Doza derived lines #12034 and #12035 possessed *Uvf-2*.

The mapping of *Uvf-2* and *Uvf-3* at higher genetic distances in Fiord/Doza#12034 and Fiord/Ac1655 $F_4$ populations, respectively from closely associated markers *KASP_Vf_0703*

and *KASP_AC×F165*, could be attributed to the robust phenotyping and genotyping data generated on F$_6$ RILs. The marker-trait association was reported in a range of crops during the past decade; the validation of such associations is often missing [45]. This is the first time markers linked with rust resistance were validated across a set of diverse genotypes of faba bean. This validation test demonstrated the value of KASP markers for MAS.

Two modern technologies, GBS-t and Illumina SNP array, were employed to identify SNPs linked with rust resistance gene *Uvf-2* and *Uvf-3*. Our assumption was that GBS-t may result in the identification of more robust markers; however, this was not the case. The SNP array is rather more straightforward and one-step ahead in comparison to GBS-t and detected better trait-marker association. The higher cost of genotyping is a bottleneck for adopting GBS-t technology in faba bean. Genotyping with 90K SNP Array [46] and DArTseq array is routinely used to map economic traits in wheat. Chhetri et al. [47] mapped stripe rust resistance gene *Yr58* in chromosome 3B using the DArTseq array and Qureshi et al. [48] developed markers linked with *Yr34* using a combination of different technologies. Both DArTseq and SNP genotyping technologies have also been used to determine marker-trait associations in durum wheat [49]. Wheat, being a major crop, attracts more research funds; hence, large sample numbers can be genotyped. However, developing large SNP arrays is not feasible in faba bean because genotyping a small sample number is economically unattractive, as compared to wheat, making GBS-t a close alternative.

Deployment of single resistance genes through conventional breeding is easy, but durability of any resistant variety carrying single resistance gene is always doubtful, especially where pathogen can evolve to acquire virulence [50]. Ijaz et al. [20] reported that the longevity of rust resistance in faba bean can be achieved when multiple resistance genes are deployed together. Combination of rust resistance genes *Uvf-2* and *Uvf-3* produced lower IT ;1$^=$ compared to IT 1C and IT 12C displayed individually by these genes. These results confirmed additivity of resistance conferred by *Uvf-2* and *Uvf-3* in the seedling stage. This finding implied the practical significance of gene pyramiding for enhanced resistance. In addition, both resistance genes *Uvf-2* and *Uvf-3* expressed an incomplete non-hypersensitive resistance as cell death remained undetected in histopathological studies that showed race-specificity in multi-pathotype tests [32]. The expression of incomplete seedling resistance is not new to faba bean because immunity to rust has never been described [13,14].

As of now, three rust resistance genes—*Uvf-1* [22], *Uvf-2*, and *Uvf-3* (present study) have been identified in faba bean. The *Uvf-1* conferring hypersensitivity against single isolate 96-Cord-2 (characterized as race-1) of *U. viciae-fabae*, was reported from Spain [22] in accession 2N52 with unknown chromosome location. The nomenclature of naming rust resistance genes *Uvf-2* and *Uvf-3* in this study was adopted from Avila et al. [22]. The faba bean genotype 2N52 is recorded in Australia under accession number Ac1653 [21]. AC1653 needs to be tested with *Uvf-2* and *Uvf-3* linked KASP markers designed in this study to recognise the uniqueness of *Uvf-1*.

The development of KASP markers for rust resistance genes *Uvf-2* and *Uvf-3* in this study has successfully filled the gap in the integration of marker technologies for faba bean breeding since the report by Avila et al. [21]. Significant achievements have been made in marker development for rust resistance in other legumes such as *Rpp1*, *Rpp2*, *Rpp3*, *Rpp4*, *Rpp5* and *Rpp6* [51,52] in soybean [*Glycine max* (L.) Merr.] against *Phakospora pachyrhizi* Syd. & P. Syd., and *Ur-3*, *Ur-4*, *Ur-5* and *Ur-9* in common bean (*Phaseolus vulgaris* L.) against *Uromyces appendiculatus* Pers. [45]. Like cereals, the examples of marker development in other legumes [45] and the present study will allow a shift of focus for rust resistance breeding in legumes from field/glasshouse based phenotypic selection to marker-assisted selection.

Rust resistance is of high importance for Australian faba bean breeders. Although, Adhikari et al. [13] and van Leur et al. [53] identified various rust resistant faba bean genotypes in Australia, precise characterisation and molecular mapping was not conducted.

The present study reported molecular markers linked with rust resistance genes *Uvf-2* and *Uvf-3* for their MAS in breeding programs. We preferred the KASP assay technology because of its user-friendly and cost-effective implementation in breeding programs. In Australia, the genes *Uvf-2* and *Uvf-3* provide resistance against a range of *U. viciae-fabae* pathotypes 0–10, 0–46, 40–31, 40–55, and 24–40 [32]. Therefore, considering the benefits of additivity and effectiveness of rust resistance against a range of pathogen isolates, both genes *Uvf-2* and *Uvf-3* should be pyramided in future cultivars through MAS to provide durable protection against faba bean rust.

**Supplementary Materials:** The following supplementary information is available online at https://www.mdpi.com/article/10.3390/agronomy11071370/s1, Table S1. Details of a set of 40 faba bean genotypes collected from six countries and used in this study; Table S2. Primer sequences of allele specific markers used for mapping of *Uvf-2* and *Uvf-*; Table S3. Linkage group (LG) statistics for Fiord/Doza#12034 RIL population forSNPs detected from RNA-Seq; Figure S1. GBS-t based linkage groups (LG) of Fiord/Doza#12034 RIL population.

**Author Contributions:** K.A. developed early generation populations; U.I. and A.S. developed recombinant inbred line populations; U.I., U.B. and H.B. conducted rust screening of mapping populations; U.I. and S.S. conducted molecular mapping; U.B. and S.K. planned and supervised molecular aspects of project; U.I. wrote manuscript; H.B., U.B., S.K. and K.A. edited the manuscript. All authors have read and agreed to the published version of the manuscript.

**Funding:** The first author was awarded the University of Sydney International Postgraduate Research Scholarship and the research was also supported by the Australian Grains Research and Development Corporation grant UA00163.

**Data Availability Statement:** Data are available from the first and the last authors.

**Conflicts of Interest:** The authors declare no conflict of interest.

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
