# Peer review of "Mapping of Two New Rust Resistance Genes Uvf-2 and Uvf-3 in Faba Bean"

_agronomy, doi:10.3390/agronomy11071370_

Round 1

Reviewer 1 Report

Based on the information in Introduction, it appears to be an important contribution. As far as gene mapping there is nothing earth shattering here, but the results appear to be of considerable practical value to bean breeders. The design is good, execution is good, presentation is OK. The presentation could perhaps be improved a bit; I had to re-read some parts three times, and keep jumping back and forth between M&M and Results, to make sure I was getting things right. If it was up to me, I would add a sentence at the end of 3.1 that the 15:1 segregation ratio clearly demonstrates that two sources of resistance carry two different unlinked resistance loci (yes, you do say it, but not explicitly enough for my taste).

Overall, I think it is a sound paper requires some minor touches (the first paragraph of the Intro is a bit disjoined; at the very least break it into two paragraphs; the second one starting with "China....").

Author Response

On behalf of all authors, I thank reviewers and editor for constructive feedback to improve this manuscript. My response to reviewers’ comments is listed below:

Based on the information in Introduction, it appears to be an important contribution. As far as gene mapping there is nothing earth shattering here, but the results appear to be of considerable practical value to bean breeders. The design is good, execution is good, presentation is OK. The presentation could perhaps be improved a bit; I had to re-read some parts three times, and keep jumping back and forth between M&M and Results, to make sure I was getting things right. If it was up to me, I would add a sentence at the end of 3.1 that the 15:1 segregation ratio clearly demonstrates that two sources of resistance carry two different unlinked resistance loci (yes, you do say it, but not explicitly enough for my taste).

Author response: Suggestions accepted, and manuscript was thoroughly revised.

Overall, I think it is a sound paper requires some minor touches (the first paragraph of the Intro is a bit disjoined; at the very least break it into two paragraphs; the second one starting with "China....").

Author response: Suggestions accepted, and manuscript was thoroughly revised.

Reviewer 2 Report

In this manuscript, Ijaz et al. report on the development of markers for mapping rust resistance genes in faba bean. The genetic materials developed are interesting, also for breeding purposes. Overall, the methods are correct and the results are clearly presented, with a good writing. However, I am suggesting below some major revision that would increase the quality of the manuscript.

A major point concerns the Sudheesh et al. 2016 reference that is often cited by the authors. Since this looks like a proceeding of a congress, I cannot find it anywhere. I think that the authors should try not referring to Sudheesh et al. 2016, but to present their results as original. Otherwise also the title may be misleading because in the present manuscript they did not map the two genes (that were indeed mapped in Sudheesh et al. 2016) but simply looked for more SNPs or for KASP assays.

Related to this point, why in Sudheesh et al. 2016 there are 16 SNPs in the q_rust_Doza region, while here only 4 of them are polymorphic? Please clarify. Similarly for Uvf-3, if the authors screened here the same population already tested in Sudheesh et al. 2016, why only three KASP out of 20 were polymorphic? In the present manuscript they have converted Illumina SNPs into KASP assays, but not mapped the Uvf-3 locus, that was already mapped in Sudheesh et al. 2016. Again, I would prefer to see all the results here and not a reference to a proceeding that I cannot find anywhere.

Another major point is related to GBS-t to map Uvf-2. It is strange that from the large set of SNPs that the authors have discovered from GBS-t, none mapped “on the other side” of Uvf-2 (I cannot say distally or proximally because it is not very clear where the centromere is..). That is, from Illumina SNPs/KASP they were able to find markers “blow” Uvf-2, while from GBS-t they only found markers “above” Uvf-2. In addition, the positions in the Joint map are identical to the positions in the single maps (Illumina/KASP and GBS-t), that leaves me with some doubts on how it was built. Did the authors put in the same input matrix for Map Manager both SNP sets? Can the authors provide as supplementary this matrix?

In the discussion the authors state that “The differences in map location could be due to the genetic architecture of mapping popula-tions used by Sudheesh et al. [35] and in the present study”, but if I have understood well, both works exploited the same mapping population.

Section 3.4 is not clear to me, since the phenotype (i.e. resistance/susceptibility) of the tested lines is not specified. Is Doza#14916 the only resistant accession? What the authors mean for “amplified false positive”, they mean that these accessions are susceptible, but they have the resistant alleles?

Other minor points:

It would be helpful for a reader not in the field to have a reference with the scale of infection types (sorry, I could not find by myself and I am not an expert of Vicia faba). Also, it would be helpful to have a picture of the ;1= infection type from the F1 between the resistant parents.

Page 2: “Doza was hetergeous” = heterogeneous?

Section 2.5: remove “(failed to amplify)”.

Section 2.6: change the format of the reference “Sudheesh et al. 2016” (also in other parts of the manuscript)

Section 3.1: in Table 2 I cannot see the results from the F1

Author Response

On behalf of all authors, I thank reviewers and editor for constructive feedback to improve this manuscript. My response to reviewers’ comments is listed below:

In this manuscript, Ijaz et al. report on the development of markers for mapping rust resistance genes in faba bean. The genetic materials developed are interesting, also for breeding purposes. Overall, the methods are correct and the results are clearly presented, with a good writing. However, I am suggesting below some major revision that would increase the quality of the manuscript.

Author response: Thanks for positive feedback. Manuscript is revised.

A major point concerns the Sudheesh et al. 2016 reference that is often cited by the authors. Since this looks like a proceeding of a congress, I cannot find it anywhere. I think that the authors should try not referring to Sudheesh et al. 2016, but to present their results as original. Otherwise also the title may be misleading because in the present manuscript they did not map the two genes (that were indeed mapped in Sudheesh et al. 2016) but simply looked for more SNPs or for KASP assays.

Author response: Text modified to address this point. Reference is taken out and the conference work is now presented in detail.

Related to this point, why in Sudheesh et al. 2016 there are 16 SNPs in the q_rust_Doza region, while here only 4 of them are polymorphic? Similarly for Uvf-3, if the authors screened here the same population already tested in Sudheesh et al. 2016, why only three KASP out of 20 were polymorphic? Please clarify.

Author response: This is a common issue in studies where QTL mapping results based on whole genome scan are used to develop kompetitive allele specific PCR markers for in-house routine use.

In the present manuscript they have converted Illumina SNPs into KASP assays, but not mapped the Uvf-3 locus, that was already mapped in Sudheesh et al. 2016. Again, I would prefer to see all the results here and not a reference to a proceeding that I cannot find anywhere.

Author response: Suggestion accepted, and manuscript modified. Sudheesh et al. (2016) reference is deleted, and data are incorporated in the revised manuscript.

Another major point is related to GBS-t to map Uvf-2. It is strange that from the large set of SNPs that the authors have discovered from GBS-t, none mapped “on the other side” of Uvf-2 (I cannot say distally or proximally because it is not very clear where the centromere is..). That is, from Illumina SNPs/KASP they were able to find markers “blow” Uvf-2, while from GBS-t they only found markers “above” Uvf-2. In addition, the positions in the Joint map are identical to the positions in the single maps (Illumina/KASP and GBS-t), that leaves me with some doubts on how it was built. Did the authors put in the same input matrix for Map Manager both SNP sets? Can the authors provide as supplementary this matrix?

Author response: Cross pollinated crops are not as molecular biology friendly and moving from F4 to F6 can change the population genetic make up through attainment of homzygosity.

In the discussion the authors state that “The differences in map location could be due to the genetic architecture of mapping populations used by Sudheesh et al. [35] and in the present study”, but if I have understood well, both works exploited the same mapping population.

 Author response: Susheesh et al. (2016) used F4 and the present study used F6 population. A high level of heterozygosity is present at the F4 stage and it can affect robustness of both phenotypic and genotypic data.

Section 3.4 is not clear to me, since the phenotype (i.e. resistance/susceptibility) of the tested lines is not specified. Is Doza#14916 the only resistant accession? What the authors mean for “amplified false positive”, they mean that these accessions are susceptible, but they have the resistant alleles?

Author response: The false positive means that the target genotype did not carry the gene based on phenotypic tests.

Other minor points:

It would be helpful for a reader not in the field to have a reference with the scale of infection types (sorry, I could not find by myself and I am not an expert of Vicia faba). Also, it would be helpful to have a picture of the ;1= infection type from the F1 between the resistant parents.

Infection type scale added. Unfortunately, we did not have F1 seed for picture.

Page 2: “Doza was hetergeous” = heterogeneous? - corrected

Section 2.5: remove “(failed to amplify)”. - removed

Section 2.6: change the format of the reference “Sudheesh et al. 2016” (also in other parts of the manuscript) – reference deleted

Section 3.1: in Table 2 I cannot see the results from the F1 – Table 2 citation removed